# GraphControl: Adding Conditional Control to Universal Graph Pre-trained Models for Graph Domain Transfer Learning

Submission Id: 682

## ABSTRACT

Graph-structured data is ubiquitous in the world which models complex relationships between objects, enabling various Web applications like paper classification, website recommendation and etc. Daily influxes of unlabeled graph data on the Web offer immense potential for these applications. Graph self-supervised algorithms have achieved significant success in acquiring generic knowledge from abundant unlabeled graph data. These pre-trained models can be applied to various downstream Web applications, saving training time and improving downstream (target) performance. However, different graphs, even across seemingly similar domains, can differ significantly in terms of attribute semantics, posing difficulties, if not infeasibility, for transferring the pre-trained models to downstream tasks. Concretely speaking, for example, the additional task-specific node information in downstream tasks (specificity) is usually deliberately omitted so that the pre-trained representation (transferability) can be leveraged. The trade-off as such is termed as "transferability-specificity dilemma" in this work. To address this challenge, we introduce an innovative deployment module coined as GraphControl, motivated by ControlNet, to realize better graph domain transfer learning. Specifically, by leveraging universal structural pre-trained models and GraphControl, we align the input space across various graphs and incorporate unique characteristics of target data as conditional inputs. These conditions will be progressively integrated into the model during fine-tuning or prompt tuning through ControlNet, facilitating personalized deployment. Extensive experiments show that our method significantly enhances the adaptability of pre-trained models on target attributed datasets, achieving 1.4-3x performance gain. Furthermore, it outperforms training-from-scratch methods on target data with a comparable margin and exhibits faster convergence.

## CCS CONCEPTS

• **Theory of computation** → **Unsupervised learning and clustering**; • **Computing methodologies** → **Transfer learning**; • **Information systems** → **Data mining**.

## KEYWORDS

Graph Neural Networks, Transfer Learning, Graph Representation Learning

**Unpublished working draft. Not for distribution.**

**ACM Reference Format:**
Anonymous Author(s). 2018. GraphControl: Adding Conditional Control to Universal Graph Pre-trained Models for Graph Domain Transfer Learning. In *Proceedings of the ACM Web Conference 2024 (WWW '24)*. ACM, New York, NY, USA, 13 pages. https://doi.org/XXXXXXX.XXXXXXX

## 1 INTRODUCTION

Graph-structured data is prevalent in Web applications, including community detection [23], website recommendation [26] and paper classification [2]. Graph representation learning plays a crucial role in these tasks, focusing on acquiring general knowledge from abundant unlabeled graph data. Recent research has explored pre-training models on such data and applying them to downstream tasks to save training time and enhance performance [29, 41, 43, 49, 53, 55, 57]. These efforts fall into two main categories.

The first group constructs training objectives based on domain-specific attributes and emphasizes pre-training and deployment on attributed graphs from the same domain [11, 38, 42]. That is, this approach requires consistent semantic meaning and feature dimensions across datasets, making it unsuitable for domain transfer. For instance, DGI [42] and MVGRL [11] are traditional self-supervised learning frameworks tailored for specific attributed graphs [6]. They are pre-trained and deployed on the same graphs. However, using these models on different attributed graphs is not feasible due to inconsistent dimensions. For example, deploying a PubMed-pretrained model on the Cora dataset [35] is unfulfillable, despite both scientific citation networks.

The second group focuses on learning transferable patterns through local structural information, enabling effective application to out-of-domain graph domains. This approach disregards node attributes during pre-training to avoid mismatches and facilitates the deployment of pre-trained models on diverse downstream datasets without relying on specific node attributes [29, 53]. For instance, GCC [29] is a graph self-supervised pre-training framework designed to capture universal topological properties across multiple graphs by using structural information as node attributes. However, during deployment, this approach does not effectively utilize downstream informative node attributes. In scenarios where nodes represent papers and contain essential information like abstracts, neglecting these attributes can impact tasks like node classification.

Nonetheless, these approaches both encounter "*transferability-specificity dilemma*":

*transferability* × − *specificity* ✓: The first group pre-trains models using domain-specific features and deploys them on the same graph, but fails to achieve domain transfer.

*transferability* ✓ − *specificity* ×: The second group aligns the feature space with structural information to achieve domain transfer, but can not effectively utilize valuable downstream node attributes.

To overcome these challenges, we propose an innovative module for effective adaptation of pre-trained models to downstream

datasets, compatible with existing pre-trained models. Specifically, we utilize universal structural pre-trained models [29] and incorporate unique features of downstream data as input conditions. Drawing inspiration from ControlNet [51], we feed structural information into the frozen pre-trained model and well-designed conditions into the trainable copy. The components are linked through zero MLPs, gradually expanding parameters from zero to incorporate valuable downstream attributes and safeguard against detrimental noise during finetuning. To ensure that the pre-trained model (trainable copy) comprehends the condition effectively, we pre-process the condition input in a manner consistent with the pre-training strategy through our condition generation module. In essence, this approach enables us to utilize the specific statistics of downstream data, leading to more effective fine-tuning or prompt tuning (*transferability* ✓ − *specificity* ✓). This innovation opens the door to more effective and efficient deployment of pre-trained models in real-world Web applications. Through extensive experiments[1], we observe that our method can enhance the adaptability of pre-trained models on downstream datasets, achieving 2-3x performance gain on Cora_ML and Amazon-Photo datasets. Furthermore, it surpasses training-from-scratch methods over 5% absolute improvement on some datasets.

Our contributions can be concluded as:

- We propose a novel deployment module coined as Graph-Control to address the "*transferability-specificity dilemma*" in graph transfer learning.
- We design a condition generation module to preprocess downstream-specific information into the pre-training data format, enabling the pre-trained model to understand the condition input effectively.
- Extensive experiments show that the proposed module significantly enhances the adaptability of pre-trained models on downstream datasets and can be seamlessly integrated with existing pre-trained models.

## 2 RELATED WORK

### 2.1 Graph Pre-training

Graph pre-training involves using existing graph data to train a generalized feature extractor. Existing self-supervised graph pre-training methods can be categorized as generative, contrastive, and predictive methods [43]. Generative methods like GAE [16], GraphMAE [13], and GraphMAE2 [12] focus on learning local relationships by reconstructing features or edges. Contrastive methods [49, 55, 57] bring positive pairs closer and push negative pairs apart to learn global relationships. Predictive methods require creating pretext tasks manually based on data statistics, like degree prediction [14], to acquire generic knowledge.

In this research endeavor, our focus lies in Graph Contrastive Learning (GCL) methods, owing to their popularity and remarkable achievements [9, 11, 38, 42, 48, 49, 55–57]. The strategies employed by GCL methods can be broadly categorized into two primary groups. The initial group formulates the training objective based on domain-specific features, exemplified by methods such as

---

[1]In this study, our focus lies on node-level downstream tasks, excluding graph classification. The alignment of node (atom) attributes in molecules (one classical data type of graph classification) mitigates the challenges in graph transfer learning.

DGI (Deep Graph Infomax) [42] and MVGRL (Contrastive Multi-View Representation Learning on Graphs) [11]. However, these approaches inherently constrain the models' generalizability to other application domains. In simpler terms, pre-trained models derived from this strategy lack the versatility to be effectively applied to attributed graphs originating from diverse application domains. Contrastingly, the second group [29, 49] directs its attention towards learning transferable patterns by discerning local graph structures, thus completely circumventing the challenge of potentially unshared attributes. Nevertheless, real-world downstream datasets are often imbued with semantic attributes. Effectively leveraging this downstream-specific information within this paradigm remains an unresolved challenge.

### 2.2 Graph Transfer Learning

Graph transfer learning [19, 28, 58] involves transferring trained model parameters to facilitate the training of new models, thereby conserving training time and occasionally enhancing downstream performance. Various strategies, such as domain adaptation [47], multi-task learning [34], and fine-tuning [37], are employed to achieve transfer learning.

In light of the remarkable achievements in pre-training techniques, this study places emphasis on fine-tuning. Initially, a generic model undergoes pre-training on extensive unlabeled graph data (source data). Subsequently, these pre-trained models are tailored for specific downstream tasks (target data). Notably, current fine-tuning methods [29] predominantly focus on adjusting pre-trained model parameters while simply incorporating target data. However, a substantial challenge arises when the feature distribution of the target data diverges from that of the source data, potentially extending to differences in feature space. For example, the pre-trained model may have a fixed input dimension (*e.g.*, 32) for structural attributes, whereas semantic attributes (*e.g.*, keywords, abstract in paper) in the target data can vary across arbitrary dimensions. Traditional fine-tuning methods inadequately tackle this issue.

To address the non-trivial problem, we propose a deployment module, coined as GraphControl, designed to incorporate downstream-specific information as input conditions. The condition will be processed to align with the format of pre-training data, enabling comprehension by pre-trained models, and steering the pre-trained model to predict more accurately, significantly enhancing the effectiveness of graph domain transfer learning.

## 3 BACKGROUND AND PROBLEM FORMULATION

In this section, we will start with the notations we use throughout the rest of the paper in Sec. 3.1. Subsequently, we will outline the specific problems under consideration in Sec. 3.2.

### 3.1 Notations

Let $\mathcal{G}, \mathcal{Y}$ represent input and label space. $f_\phi(\cdot)$ represents graph predictor which consists of a GNN encoder $g_\theta(\cdot)$ and a classifier $p_\omega(\cdot)$. The graph predictor $f_\phi : \mathcal{G} \mapsto \mathcal{Y}$ maps instance $G = (A, X, Y) \in \mathcal{G}$ to label $Y \in \mathcal{Y}$ where $A \in \mathbb{R}^{N \times N}$ is the adjacency matrix and

$X \in \mathbb{R}^{N \times d}$ is the node attribute matrix. Here, $N$, $d$ denote the number of nodes and attributes, respectively. Let $G_i$ denote a subgraph centered around node $i$ sampled from the original graph $G$.

## 3.2 Problem Definition

### 3.2.1 Universal Graph Representation Learning, UGRL.
UGRL endeavors to acquire a universal feature extractor $g_\theta$ from abundant unlabeled graph data, encapsulating common and generic knowledge. This extractor is versatile and applicable to diverse datasets sourced from similar domains. Varying and occasionally absent node attributes pose challenges for effective transfer in node attribute-based pre-training. To address this challenge, we introduce structure pre-training models that mainly utilize structural information, collectively termed UGRL in this paper. These models offer a solution, ensuring efficient knowledge transfer despite disparities in node attributes across datasets.

GCC[29] is a classical structural pre-training method that leverages structural information as input. This approach aligns the input space across all datasets using structural information, facilitating domain transfer. To learn common knowledge, GCC will sample subgraphs $\{G_i\}_{i=1}^{N}$ from the original large graph $G$ and embed subgraphs with similar local structures closely through subgraph instance discrimination. Inspired by GCC, we employ generalized positional embedding as input features during pre-training. Formally, given an adjacency matrix $A$ and the corresponding degree matrix $D$, we conduct eigen-decomposition on its normalized graph Laplacian $U\Lambda U^T = I - D^{-\frac{1}{2}}AD^{-\frac{1}{2}}$. The top eigenvectors in $U$ will serve as generalized positional embedding. The GNN encoder $g_\theta : \mathbb{R}^{N \times k} \mapsto \mathbb{R}^{N \times l}$ maps the positional embedding $P \in \mathbb{R}^{N \times k}$ ($k$ set as 32 in this paper) to node embedding $H \in \mathbb{R}^{N \times l}$. To learn transferable structural patterns from positional embedding, we will maximize the mutual information between two similar subgraphs. Taking the InfoNCE loss[27] as an example, the formulation follows:

$$
\begin{aligned}
\mathcal{L}_{\text{MI}}(g_\theta; G) = & -\underset{G_i, G_i' \in G}{\mathbb{E}} \left\| g_\theta(G_i) - g_\theta(G_i') \right\|^2 \\
& + \underset{G_i \in G}{\mathbb{E}} \log \underset{G_j \in G}{\mathbb{E}} \left[ e^{\left\| g_\theta(G_i) - g_\theta(G_j) \right\|^2} \right],
\end{aligned}
\tag{1}
$$

where $g_\theta$ denotes GNN encoder with readout function, $G_i$, $G_i'$ represents subgraphs centered around node $i$ sampled from the original graph $G$. The sampling strategy is random walk with restart which is also adopted in GCC [29] and RoSA [55]. So this method is scalable on large graphs. $G_i$ and $G_i'$ share similar local structural information, serving as positive pairs, while $G_i$ and $G_j$ (sampled from different central nodes) act as negative samples. Through this self-supervised objective, UGRL obtains pre-trained models applicable to various downstream datasets, addressing specific tasks like node classification. This learning framework is commonly referred to as graph transfer learning.

### 3.2.2 Graph Transfer Learning.
Graph transfer learning aims to leverage the universal knowledge within a pre-trained model, trained on source data, and apply it to target data for specific tasks. There exists source data $\mathcal{D}^{\text{source}}$ and target data $\mathcal{D}^{\text{target}}$ from similar domains. In this paper, we assume source data is abundant but without labels and target data is limited but with labels. UGRL

acquires pre-trained models $g_\theta^\star$ on the source data, which are then fine-tuned on the limited target data to accomplish specific tasks.

Take the downstream node classification tasks as an example, it involves learning a conditional probability $P(Y \mid G; \phi)$ to categorize unlabeled nodes. To model this probability, the graph predictor $f_\phi(\cdot) = p_\omega \circ g_\theta^\star(\cdot)$ is employed where $p_\omega$ is a classifier and $g_\theta^\star$ is pre-trained GNN encoder obtained by the last part. The graph predictor is then optimized with training nodes from target data:

$$
f_\phi^* = \underset{\phi}{\arg\min} \, \mathbb{E}_{G \sim \mathcal{D}^{\text{target}}} \ell \left( f_\phi \left( X_{\text{train}}, A \right), Y_{\text{train}} \right),
\tag{2}
$$

where $X_{\text{train}}$ represents the attribute set of training nodes and $Y_{\text{train}}$ denotes their labels, and $\ell(\cdot, \cdot)$ is cross-entropy loss. Finally, the optimal graph predictor $f_\phi^*$ is utilized for classifying testing nodes.

However, during pre-training, we only utilize structural information to obtain transferable pre-trained models while disregarding non-transferable node attributes. Sometimes, downstream data includes specific node attributes (*e.g.*, age, gender, and interests) crucial for the task but incompatible with the pre-trained model (due to disparities in feature space and dimensions). Incorporating these meaningful attributes into the model and guiding it towards superior performance represents a substantial challenge. We will present the solution to this challenge in Sec. 4.

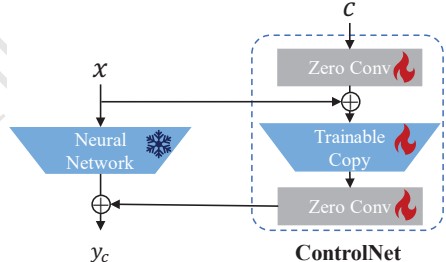

**Figure 1: ControlNet injects conditions into neural network. $x$ represents original input and $c$ denotes condition input.**

## 4 GRAPHCONTROL: GRAPH TRANSFER LEARNING WITH CONDITIONAL CONTROL

In this section, we will outline our approach to address the aforementioned challenge. Firstly, we retrospect the ControlNet in Sec. 4.1. Then in Sec. 4.2, we first introduce the common challenges of adding conditional control to the graph data in Sec. 4.2.1, followed up by the condition generation mechanism proposed in Sec. 4.2.2. Subsequently, we detail our module GraphControl designed to adapt the pre-trained model to the target data in Sec. 4.2.3. Lastly, we demonstrate how to incorporate our module with fine-tuning and prompt-tuning techniques in Sec. 4.3. Furthermore, we add time complexity analysis in Appendix B due to the space limit.

### 4.1 Background: ControlNet

Firstly, let us retrospect the concepts of ControlNet [51]. ControlNet is a neural network architecture designed to incorporate spatial conditioning controls into pre-trained diffusion models [5, 15, 33] to generate customized images. Specifically, it freezes the pre-trained model and reuses the deep and robust encoding layers as a robust

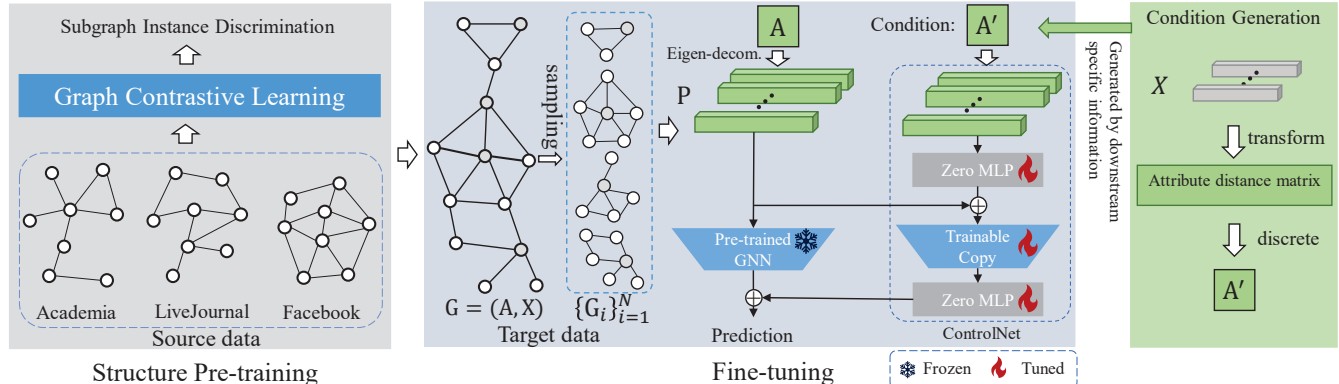

**Figure 2: The pipeline of Graph Domain Transfer Learning with GraphControl: Universal structure pre-training will be applied on extensive source data, then the pre-trained model will be deployed on target data with GraphControl , which includes Condition Generation and modified ControlNet.**

backbone (trainable copy) for acquiring diverse conditional controls. The trainable copy and original model are linked by zero convolution layers, progressively growing parameters from zero, ensuring a noise-free fine-tuning process [51]. This approach allows us to control diffusion models with learned conditions. For example, utilizing a human pose stretch as a condition ensures that all generated images share the same pose [51].

## 4.2 GraphControl: Transfer Learning with ControlNet

*4.2.1 Leveraging Node Attributes in Graph Transfer Learning.* In graph domain, the downstream-specific node attributes pose compatibility challenges with the pre-trained model, primarily due to the disparities in terms of feature semantics and dimensions. One straightforward approach is to train a dedicated feature extractor for these attributes and integrate it with pre-trained models for prediction. However, this solution encounters three main issues: (i) The downstream data is of a small scale, akin to few-shot scenarios [8], making training-from-scratch susceptible to overfitting on the training data and poor generalization on testing data. (ii) The pre-trained model's assistance remains limited, failing to fully leverage its potential. (iii) Selecting an appropriate feature extractor for node attributes is an open question, as different datasets may require different extractors. A brute-force approach trains with all choices and selects the best, incurring high training costs [46].

Our purpose is to enable the existing structural knowledge pre-training framework to utilize the node attributes of different downstream datasets in the fine-tuning or prompt-tuning stage. To achieve this, we draw inspiration from ControlNet [51], a neural network architecture that incorporates conditioning controls into large pre-trained text-to-image diffusion models. Considering the unique structure of the graph data, there are two primary distinctions that set our work apart from ControlNet:

- The motivation and application domain: ControlNet aims to generate customized images through user instructions. In this study, we address the challenge of graph domain transfer learning by incorporating elaborate conditional

control, generated from downstream specific information, into universal pre-trained models.
- The input condition: In ControlNet, the input condition for pre-trained text-to-image models is easily designed using sketches (*e.g.*, cartoon line drawings, shape normals). In our study, we utilize structural pre-trained models for graph transfer learning. Integrating downstream-specific information as a comprehensible condition for pre-trained models is non-trivial.

*4.2.2 Condition Generation in the Graph Domain.* In order to obtain conditions meeting the requirements in the second distinction, we propose a condition generation module depicted in Figure 2 (green region). It utilizes the downstream-specific characteristics like node attributes to design the condition in a similar format to the adjacent matrix. Specifically, firstly, we measure the distance between nodes through a kernel function $\kappa(\cdot, \cdot)$. Thus we have a kernel matrix [18] (attribute distance matrix) $K \in \mathbb{R}^{N \times N}$ where $K_{ij} = \kappa(x_i, x_j)$. In this work, we use the linear kernel with normalized term (cosine similarity function) for computational simplicity:

$$\kappa(x_i, x_j) = \frac{x_i^T x_j}{\|x_i\| \|x_j\|}. \tag{3}$$

We then discretize this kernel matrix by applying a threshold filter $v$ to it. The values that are bigger than the threshold $v$ will be set to 1 otherwise 0:

$$A'_{i,j} = \begin{cases} 0, & \text{if } K_{i,j} \leq v \\ 1, & \text{o.w.} \end{cases} \tag{4}$$

We call $A'$ as feature adjacent matrix that aligns and maps node features of different dimensions and different semantics to the adjacency matrix space. Finally, we will use the same process during pre-training to obtain the generalized positional embedding $P'$, which will be used during fine-tuning. For non-attributed downstream graphs lacking node attributes, node embedding strategies like Node2Vec [10] can be applied to generate node attributes. These attributes can then be utilized for creating conditions via our condition generation module.

Next, we elucidate the integration of ControlNet into the graph domain, leveraging our designed condition to facilitate graph domain transfer.

*4.2.3   Overall Framework of GraphControl .*  In this work, we draw inspiration from ControNet to solve the challenges of graph domain transfer learning. Considering the non-euclidean nature and oversmoothing problem [4] in graph domain, we substitute zero convolution layers with zero MLPs rather than zero graph convolution layers. We leverage universal structural pre-trained models and incorporate the downstream-specific information as condition input, effectively tackling the "transferability-specificity dilemma." The structural information of target data will be fed to the frozen pre-trained model (to avoid catastrophic forgetting [17, 20, 24, 25]) and the elaborate condition (generated by Condition Generation Module) will be fed into the trainable copy. These two components are linked by zero MLPs, gradually growing the parameters from zero. This approach ensures that no harmful noise affects the fine-tuning process while progressively incorporating downstream-specific information.

The procedure of our method can be formalized as follows:

$$H_c = g_\theta^\star(P) + \mathcal{Z}_2(g_c(P + \mathcal{Z}_1(P'))), \tag{5}$$

where $\mathcal{Z}_1$ and $\mathcal{Z}_2$ represent the first and the second zero MLP, and $g_c(\cdot)$ represents the trainable copy of the pre-trained encoder $g_\theta^\star$. Similar to ControlNet, because the parameters of the zero MLP are set to 0 during initialization, we have $\mathcal{Z}_2(g_c(P + \mathcal{Z}_1(P_c))) = 0$. Hence, during initialization, our model's output aligns with using the pre-trained encoder alone. Throughout optimization, the downstream-specific information is progressively integrated.

## 4.3   GraphControl in Two Learning Scenarios

*4.3.1   Fine-tuning with GraphControl.*  Given an input graph $G = (A, X)$, our process begins with preprocessing the graph data, involving subgraph sampling, generalized positional embedding calculation, and condition generation. In the training phase, we keep the parameters of the pre-trained GNN encoder $g_\theta$ fixed to prevent catastrophic forgetting. The original positional embedding $P$ is input into $g_\theta^\star$, and the generated condition is input into the ControlNet. The resulting representations $H_c$ are utilized for specific tasks. For example, in the node classification task, a linear classifier is added to map these representations to predicted labels. The classification error is then calculated using the cross-entropy loss function. All components are optimized in an end-to-end manner. The entire procedure is outlined in Algorithm 1.

*4.3.2   Graph Prompt Tuning with GraphControl.*  In the last section, we introduce how to perform fine-tuning with our method on target data. In scenarios where training data for the target dataset is notably scarce (*e.g.*, fewshot setting), tuning all parameters can result in overfitting and difficulties in generalizing effectively on the test set. To address these challenges, graph prompt tuning methods [7, 21, 39, 40, 54] have emerged which focus on tuning only a few parameters of the prompt. In this section, we will demonstrate that our method (GraphControl) can seamlessly integrate with existing graph prompt methods, significantly enhancing downstream performance. Taking the GPF graph prompt tuning method [7] as an example, the workflow is illustrated in Figure 3. Firstly, two

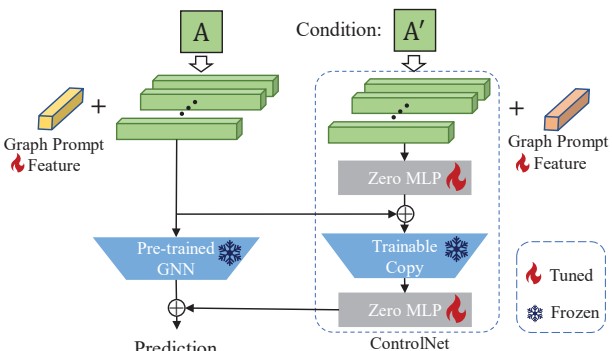

**Figure 3: Graph prompt tuning with GraphControl.**

trainable graph prompt features $p, p' \in \mathbb{R}^{1 \times k}$ are randomly initialized. Then, these prompt features are broadcasted to be added to the input features. The formulation is as follows:

$$H_c = g_\theta^\star(P + q) + \mathcal{Z}_2(g_c((P + q) + \mathcal{Z}_1(P' + q'))), \tag{6}$$

In contrast to the previous section, in graph prompt tuning, the parameters of the trainable copy $g_c$ will be frozen to prevent overfitting. Besides, more intricate graph prompt methods, such as All-in-One (ProG) [40], can be integrated with our method to enhance downstream performance. Detailed experiments are in Sec. 5.3.2.

## 5   EXPERIMENTS

In this section, we first introduce the datasets, baselines, and experimental setup in Sec. 5.1, 5.2 and 5.3 respectively. Secondly, we conduct main experiments under fine-tuning (Sec. 5.3.1) and prompt tuning (Sec. 5.3.2) to prove the effectiveness of GraphControl. We then perform an ablation study to demonstrate the effectiveness of each proposed component in Sec. 5.4. Lastly, we analyze the convergence of GraphControl (Sec.5.5) and the impact of important hyper-parameters (Sec. 5.6).

## 5.1   Datasets

**Pre-training datasets.**    The details of the pre-training datasets are presented in Table 6 and Appendix C.1. Notably, these datasets are substantial in scale, with the largest graph (LiveJournal [1]) comprising approximately 4.8 million nodes and 85 million edges.
**Downstream datasets.**    We select eight public benchmark datasets as target data that include four attributed datasets (*i.e.*, Cora_ML, Amazon-Photo, DBLP, and Coauthor-Physics), and four non-attributed datasets (*i.e.*, USA-Airport, Europe-Airport, Brazil-Airport, and H-index) to evaluate the effectiveness of GraphControl. The statistics of datasets is in Table 1. Detailed illustrations of these datasets can be found in Appendix C.1.

## 5.2   Baselines

We evaluate GraphControl with four self-supervised pre-training methods (using GIN as encoder): Deep Graph Contrastive Representation Learning (**GRACE**) [57], A Simple Framework for Graph Contrastive Learning without Data Augmentation (**simGRACE**) [44],

**Table 1: Statistics of datasets. These datasets can be further classified into attributed graphs and non-attributed graphs.**

|  | #Nodes | #Edges | #Attributes | #Classes |
|---|---|---|---|---|
| Cora_ML[3] | 2,995 | 16,316 | 2,879 | 7 |
| Amazon-Photo[36] | 7,650 | 238,162 | 745 | 8 |
| DBLP[3] | 17,716 | 105,734 | 1,639 | 4 |
| Coauthor-Physics[36] | 34,493 | 495,924 | 8,415 | 5 |
| USA-Airport[30] | 1,190 | 27,198 | - | 4 |
| Europe-Airport[30] | 399 | 5,995 | - | 4 |
| Brazil-Airport[30] | 131 | 1,047 | - | 4 |
| H-index[29] | 5,000 | 44,020 | - | 44 |

Covariance-Preserving Feature Augmentation for Graph Contrastive Learning (**COSTA**) [52], and A Robust Self-Aligned Framework for Node-Node Graph Contrastive Learning (**RoSA**) [55]. Detailed descriptions of these methods are available in Appendix C.3. Except for GCC, other pre-training methods are designed for attributed graphs. To integrate them into our setting, we replace their input with structural information, disregarding the original node attributes during pre-training. To demonstrate the superiority of our approach over training from scratch, we compare it with the supervised GIN model, initialized randomly and trained on target data. Notably, GCC's encoder is based on GIN but with a little different implementation (*e.g.*, incorporates additional information as input, further details are provided in Appendix). GCC(rand) signifies the utilization of a randomly initialized GCC encoder, trained from scratch on the target data. Additionally, we include two baselines that only use node attributes (*i.e.*, MLP [31]) and structural information (*i.e.*, Node2Vec [10]) of downstream datasets to demonstrate the effectiveness of both for classification. Considering the abundance of source data and its occasional unavailability (we only have access to pre-trained models), domain adaptation baselines [47] are not included in this work.

## 5.3 Graph Domain Transfer Learning

*5.3.1 Fine-tuning with GraphControl.* In this subsection, we evaluate the effectiveness of GraphControl on target data by fine-tuning.
**Experimental setup.** For pre-trained models, GCC [29] is pre-trained on abundant unlabeled large graphs (*e.g.*, Facebook [32], LiveJournal [1]), we use their released pre-trained checkpoint[2]. In the case of GRACE, simGRACE, RoSA, and COSTA, we perform pre-training on the downstream graphs, excluding node attributes. During fine-tuning, we incorporate the original node attributes. All pre-training methods use a 4-layer Graph Isomorphism Network (GIN) [45] with 64 hidden units as encoders.

Regarding data splitting, we randomly divide the training and testing data into a 1:9 ratio, and the results represent the mean accuracy with a standard deviation of 20 runs with different random seeds. Details and hyperparameters can be found in Appendix C.2.
**Analysis.** From Table 2, we can draw the following conclusions: firstly, structural pre-training methods can learn transferable structural patterns because GCC surpasses GCC(rand)[3] with comparable

---

[2]https://github.com/THUDM/GCC
[3]GCC(rand) refers to a randomly initialized encoder of GCC, trained from scratch on target data, focusing on structural information.

margins. For instance, on H-index and Cora_ML datasets, GCC achieves over 5% absolute improvement compared to GCC(rand).

Secondly, applying structural pre-training methods directly to target attributed graphs fails to achieve satisfactory performance and notably lags behind training-from-scratch methods (*e.g.*, GIN(A,X)) on target data. This underscores the essential role of downstream-specific information (*e.g.*, node attributes) for optimal performance. For instance, on the DBLP dataset, GCC achieves only 57% accuracy, lagging behind GIN(A,X) by approximately 17%.

Thirdly, deploying structural pre-trained models on target data with GraphControl significantly enhances performance. For instance, on Cora_ML and Photo datasets, our method achieves 2-3x performance gains compared to direct deployment. Moreover, when pre-trained models are combined with GraphControl , intelligently leveraging downstream-specific information, they outperform training-from-scratch methods on target data, showcasing GraphControl ' ability to fully harness the potential of pre-trained models. Even for non-attributed target data, our method can enhance downstream performance with additional node embeddings from Node2Vec [10]. Specifically, GRACE with GraphControl outperforms GIN(A,$X_{PE}$) by approximately 5% absolute improvement.

These statistics show the effectiveness of our module for deploying universal pre-trained models on target data.

*5.3.2 Prompt Tuning with GraphControl (Few-shot classification).* In many real-world scenarios, the target data is notably limited, with only a few training samples for each class. Few-shot learning is a well-known case of low-resource scenarios. Standard fine-tuning tends to overfit on the training data, leading to poor generalization. To solve these problems, Graph prompt tuning emerged which can align the training objectives and train a few parameters of prompt. In this section, we will perform experiments of existing graph prompt tuning with GraphControl under few-shot setting.
**Baselines & Experimental setup.** We choose two graph prompt methods, GPF [7] and ProG [40], which are not limited to specific pre-trained GNN models. Other graph prompt methods like GPPT [39], GraphPrompt [21], and SGL-PT [54] heavily rely on specific pre-trained models will not included in this study. GPF introduces trainable graph prompt features applied to the original graph, imitating any graph manipulations. ProG is a more complex version, inserting a prompt graph comprising multiple prompt features and relations into the original graph. For the pre-trained model, we adopt GCC here for simplicity.

'Finetuned GIN' and 'Finetuned GCC' refer to randomly initialized GIN and pre-trained GCC fine-tuned on target data. 'GCC+GPF' indicates pre-trained GCC prompt tuning on target data with GPF, while 'Ours+GPF' involves deploying pre-trained GCC with GraphControl using GPF as prompt tuning. 'GCC+PorG' and 'Ours+PorG' use ProG as prompt tuning method.

As for data splitting, target data is initially divided into 1:9 for candidate and testing data. In 3-shot(5-shot) setting, 3(5) samples per class are subsequently selected from candidate data for training. Results show mean accuracy with standard deviation over 20 different random seeds. For more details, please refer to Appendix C.2.
**Analysis.** From Table 3, we can draw the following conclusions: firstly, we can see the 'Finetuned GIN' achieves the worst performance because the training from scratch will easily overfit on

**Table 2: Experimental results of baselines and our method on downstream datasets. In the data column, $A$ represents adjacent matrix, $X$ denotes node attribute matrix and $X_{PE}$ means positional embeddings. Rows with gray background denote our method.**

| Data | Methods | Cora_ML | Photo | DBLP | Physics | USA | Europe | Brazil | H-index |
|---|---|---|---|---|---|---|---|---|---|
| X | MLP[31] | 60.31±2.96 | 77.56±2.42 | 64.47±1.36 | 88.90±1.10 | - | - | - | - |
| A | Node2Vec[10] | 69.93±1.27 | 84.08±0.63 | 77.52±0.38 | 88.13±0.39 | 59.59±2.04 | 47.92±3.66 | 46.53±8.41 | 75.02±0.50 |
| A,$X_{PE}$ | GIN[45] | 29.94±1.37 | 30.41±1.07 | 57.53±0.78 | 54.76±0.69 | 56.33±1.90 | 49.72±3.05 | 57.63±8.96 | 69.90±1.26 |
| A,X | GIN[45] | 69.57±3.65 | 79.71±4.72 | 74.62±3.00 | 92.02±2.79 | 58.89±2.70 | 47.85±4.86 | 58.52±9.98 | 72.23±1.20 |
| A,$X_{PE}$ | GCC(rand) | 26.34±1.40 | 26.15±1.20 | 53.46±0.79 | 54.30±0.68 | 54.85±2.31 | 42.60±3.31 | 51.20±8.49 | 64.18±1.83 |
| A,$X_{PE}$ | GCC[29] | 31.14±1.23 | 33.85±1.19 | 57.02±0.68 | 56.25±0.37 | 55.80±2.23 | 47.35±3.44 | 57.92±9.00 | 70.31±1.89 |
| A,$X_{PE}$,X | +GraphControl | 77.43±1.62 | 88.65±0.60 | 80.25±0.90 | 94.31±0.12 | 57.03±2.21 | 50.53±3.43 | 59.28±8.14 | 73.55±0.70 |
| A,$X_{PE}$ | GRACE[57] | 30.74±1.48 | 32.64±1.57 | 58.43±0.37 | 59.86±1.96 | 57.68±1.75 | 50.49±2.90 | 57.98±9.45 | 69.68±2.18 |
| A,$X_{PE}$,X | +GraphControl | 77.26±1.50 | 88.78±0.61 | 80.42±0.65 | 94.12±0.24 | 58.94±1.84 | 52.83±3.10 | 59.92±7.59 | 74.47±0.07 |
| A,$X_{PE}$ | simGRACE[44] | 30.39±1.82 | 33.62±1.52 | 57.87±0.32 | 59.82±2.93 | 57.11±1.90 | 50.22±3.91 | 58.09±8.50 | 69.65±1.50 |
| A,$X_{PE}$,X | +GraphControl | 77.34±1.08 | 89.66±0.56 | 80.33±0.69 | 94.03±0.47 | 59.40±1.62 | 51.15±3.17 | 59.41±7.66 | 76.10±0.70 |
| A,$X_{PE}$ | RoSA[55] | 30.96±0.81 | 33.42±1.59 | 56.41±0.70 | 60.14±2.48 | 57.18±2.02 | 50.32±3.78 | 58.99±8.30 | 69.80±2.48 |
| A,$X_{PE}$,X | +GraphControl | 77.40±1.06 | 89.35±0.61 | 80.23±0.79 | 94.22±0.26 | 58.71±1.35 | 51.89±2.69 | 59.16±7.13 | 74.22±1.46 |
| A,$X_{PE}$ | COSTA[52] | 30.07±1.31 | 33.22±1.28 | 59.01±0.19 | 59.96±3.29 | 57.07±2.53 | 50.33±3.64 | 59.55±9.30 | 68.49±2.10 |
| A,$X_{PE}$,X | +GraphControl | 76.63±1.67 | 89.17±1.14 | 80.74±0.65 | 94.02±0.31 | 59.00±1.82 | 51.88±3.08 | 62.16±6.95 | 73.57±2.17 |

**Table 3: Experimental results of fine-tuning (FT) and prompt tuning (PT) under few-shot settings (3-shot and 5-shot).**

| | | USA | | Europe | |
|---|---|---|---|---|---|
| | | 3-shot | 5-shot | 3-shot | 5-shot |
| FT | Finetuned GIN | 34.28±4.06 | 35.73±4.45 | 37.99±4.38 | 40.61±3.18 |
| | Finetuned GCC | 48.75±4.76 | 51.76±4.98 | 45.08±4.24 | 48.62±3.77 |
| PT | GCC+GPF[7] | 49.10±4.70 | 50.78±5.19 | 47.10±3.57 | 49.69±3.47 |
| | Ours+GPF[7] | 50.40±3.33 | 53.05±4.52 | 47.50±3.99 | 50.29±2.42 |
| | GCC+ProG[40] | 48.80±4.36 | 49.36±5.64 | 46.04±4.38 | 48.47±3.67 |
| | Ours+ProG[40] | 49.73±4.34 | 52.61±5.22 | 46.81±4.39 | 50.65±2.93 |

limited training data. 'Finetuned GCC' achieves a decent result but lags behind prompt methods with few training samples. For example, under 3-shot, 'Finetuned GCC' is outperformed by 'GCC+GPF' and 'GCC+ProG', but surpasses them with more training data (5-shot). This highlights the greater effectiveness of prompt methods under limited resources.

Secondly, with prompt tuning, our method GraphControl can still enhance the downstream performance. Specifically, both 'Ours+ProG' and 'Ours+GPF' outperform their corresponding baselines ('GCC+ProG' and 'GCC+GPF') by 2% absolute improvement. In the small-scale dataset like 5-shot Europe-Airport, 'Ours+ProG' reaches comparable performance to full-shot in the last section.

## 5.4 Ablation Studies

In this section, we assess the effectiveness of GraphControl by masking each component. Ours (soft C) uses soft attribute distance matrix $K$ as condition. Ours (w/o zero) removes zero MLPs in ControlNet. Ours (w/o CG) removes the condition generation and ControlNet, similar to finetuning GCC. Ours (w/o frozen pre.) excludes the frozen pre-trained model branch, utilizing only the ControlNet branch. Lastly, 'Simple Cat.' signifies a basic approach: training a dedicated feature extractor for downstream attributes from scratch

and integrating it with pre-trained models for predictions. As for the pre-trained model, we use GCC in this section.

Each component is crucial for the effectiveness of the method, as shown in Table 4. Specifically, Ours(w/o CG) performs poorly on attributed datasets, emphasizing the importance of downstream-specific information. Ours(soft C) also underperforms, highlighting the significance of aligning the format of condition and input during pre-training. Ours(w/o zero) lags behind GraphControl by a comparable margin, indicating the importance of zero MLPs in linking the frozen pre-trained model and the trainable copy, avoiding detrimental noise during fine-tuning. Ours(w/o frozen pre.) is also inferior to GraphControl, indicating the effectiveness of incorporating common knowledge from the pre-trained model. Finally, 'Simply Cat.' achieves subpar results on most datasets, emphasizing the risks of overfitting when training from scratch on limited data, especially in smaller datasets like Cora_ML (10% lower than GraphControl).

## 5.5 Convergence Analysis

Earlier sections demonstrate the efficacy of our method in performance enhancement. In this section, we delve into the convergence speed analysis of GraphControl. Here, 'GIN' denotes training from scratch, while GraphControl signifies using GCC as the pre-trained model and fine-tuning it with GraphControl.

As depicted in Figure 4, GraphControl achieves convergence within 100 epochs on all datasets, whereas GIN reaches the best performance around 600 epochs on Cora_ML and Photo datasets, exhibiting instability. Our approach not only improves performance but also notably reduces training time in downstream applications.

## 5.6 Sensitivity Analysis

In this section, we analyze crucial hyperparameters, starting with the impact of the threshold used in condition generation, followed by an analysis of the subsampling hyperparameters.

**Table 4: Ablation studies for GraphControl by masking out each component. The bold and underlined results means the top-1 performance and the underline represents the second performance.**

| Data | Methods | Cora_ML | Photo | DBLP | Physics | USA | Europe | Brazil | H-index |
|---|---|---|---|---|---|---|---|---|---|
| A,$X_{PE}$,X | Ours | **77.43±1.62** | **88.65±0.60** | **80.25±0.90** | **94.31±0.12** | **57.03±2.21** | **50.53±3.43** | **59.28±8.14** | **73.55±0.70** |
| A,$X_{PE}$,X | Ours(soft C) | 27.42±3.32 | 51.85±7.30 | 51.42±3.37 | 77.75±2.51 | 55.03±2.57 | 48.67±3.95 | 58.81±8.31 | 73.26±0.63 |
| A,$X_{PE}$,X | Ours(w/o zero) | 71.93±2.73 | 81.30±4.33 | *77.24±5.55* | 93.70±0.25 | 55.35±3.08 | 48.49±3.39 | 58.03±9.91 | 71.71±1.63 |
| A,$X_{PE}$ | Ours(w/o CG) | 20.29±2.09 | 27.31±3.09 | 49.62±3.54 | 51.96±0.70 | 55.31±2.32 | 47.93±3.41 | 56.65±8.76 | 72.79±0.53 |
| A,X | Ours(w/o frozen pre.) | 75.06±1.90 | 82.38±4.72 | 79.22±1.58 | 93.56±0.22 | 55.85±2.71 | 47.75±3.04 | 56.57±8.65 | 73.16±0.72 |
| A,$X_{PE}$,X | Simple Cat. | 66.64±1.41 | 82.80±0.95 | 67.32±0.60 | 90.56±0.46 | 53.59±4.74 | 45.28±5.29 | 57.67±7.14 | 72.74±1.16 |

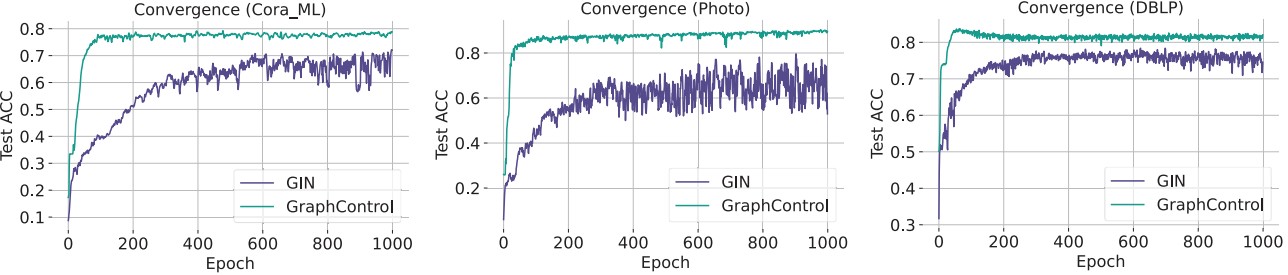

**Figure 4: The convergence analysis on GIN and GraphControl.**

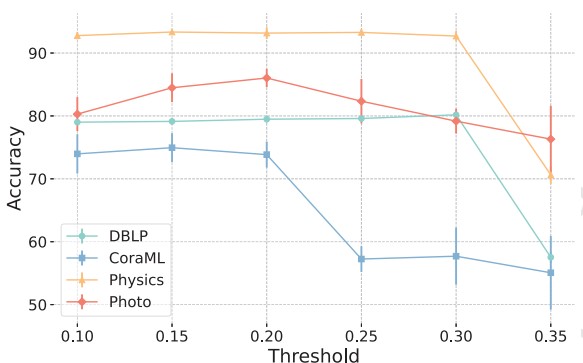

**Figure 5: Sensitivity analysis on threshold.**

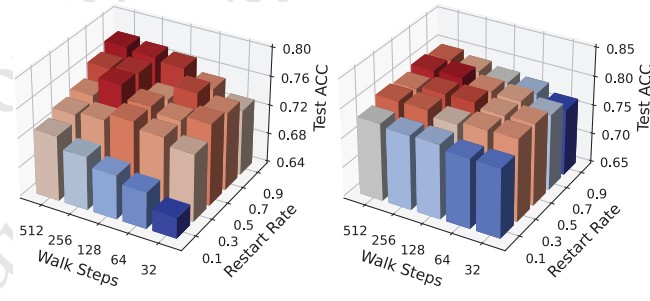

**Figure 6: Analysis of subsampling hyperparameters on Cora_ML (left) and DBLP (right) datasets.**

*5.6.1 Analysis on threshold for discretization.* In the process of condition generation, we discretize the attribute distance matrix using a specific threshold, converting it into a feature-based adjacency matrix to align with the input space during pre-training. Empirically, we explore the impact of this threshold, ranging from 0.1 to 0.35. The results, shown in Figure 5, indicate stable performance from 0.1 to 0.2 on DBLP, Cora_ML, and Physics datasets. However, when the threshold exceeds 0.3, most datasets experience a rapid drop in performance due to the matrix becoming overly sparse and providing limited information. Optimal thresholds range from 0.15 to 0.2, guiding our experiments across most datasets.

*5.6.2 Analysis on hyper-parameters of subsampling.* In this work, random walk with restart serves as the subsampling technique, with walk steps and restart rate as pivotal hyperparameters. Walk steps are selected from {32, 64, 128, 256, 512}, and the restart rate spans {0.1, 0.3, 0.5, 0.7, 0.9}. Based on Figure 6, optimal results are observed with 256 and 512 walk steps, alongside restart rates of 0.7

and 0.9. For memory efficiency, we standardize walk steps to 256 across all datasets and set the restart rate to 0.8 for most datasets.

## 6 CONCLUSION

In this work, we propose a novel deployment module coined as GraphControl to address the challenges of the 'pre-training and finetuning (or prompt-tuning)' paradigm in graph domain transfer learning. GraphControl seamlessly integrates with existing universal structural pre-trained models, significantly boosting their performance on target data by intelligently incorporating downstream-specific information. Specifically, to achieve this, we draw inspiration from ControlNet and apply its core concepts to graph domain transfer learning. Downstream-specific information is processed into conditions using our condition generation module and gradually integrated for enhanced performance. Extensive experiments on diverse real-world datasets demonstrate the superiority of Graph-Control in fine-tuning and prompt tuning scenarios, substantially improving the adaptability of pre-trained models on target data.

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

---

**Algorithm 1:** GraphControl algorithm

**Input:** Frozen pre-trained GNN encoder $g_\theta^\star$, trainable copy of pre-trained model $g_{\theta_c}$, two zero MLPs $\mathcal{Z}_1, \mathcal{Z}_2$ with parameters $\theta_{Z_1}, \theta_{Z_2}$, random initialized linear classifier $p_\omega$, input graph $G = (A, X, Y)$, sampler function $\mathcal{T}(G, i)$, training epochs $E$, learning rate $\eta$.

**Output:** Optimized models, $g_{\theta_c}, \mathcal{Z}_1, \mathcal{Z}_2, p_\omega$

/* subsampling                                           */

**for** $i \leftarrow 1$ **to** $N$ **do**
  $G_i = \mathcal{T}(G, i) = (A_i, X_i)$;
  Generate condition $A_i'$ through Condition Generation
    Module using node attributes;
  Generate positional embedding $P_i$ from graph adjacency
    matrix $A_i$ and conditional positional embedding $P_i'$
    from attribute adjacency matrix $A_i'$;
  $G_i = (A_i, P_i, P_i', y_i)$
**end**

$\mathcal{S} = \{G_i\}, i = 1, ..., N$ ;     // collect training samples

**for** $e \leftarrow 1$ **to** $E$ **do**
  Sampled batch $\mathcal{B} = \{G_i\}_{i=1}^B \in \mathcal{S}$;
  /* For symbol unclutter, we omit subscript */
  Batched graph $G = (A, P, P', Y)$;
  /* Forward                                           */
  $H \leftarrow g_\theta(P) + \mathcal{Z}_2(g_{\theta_c}(P + \mathcal{Z}_1(P')))$ ;
  $\ell_{\text{sup}} \leftarrow \mathcal{L}(p_\omega(H), Y)$ ;
  /* Backward                                          */
  $\theta_c \leftarrow \theta_c - \eta \nabla_{\theta_c} \ell_{\text{sup}}; \theta_{Z_1} \leftarrow \theta_{Z_1} - \eta \nabla_{\theta_{Z_1}} \ell_{\text{sup}}$;
    $\theta_{Z_2} \leftarrow \theta_{Z_2} - \eta \nabla_{\theta_{Z_2}} \ell_{\text{sup}}$ ;
**end**

## A  ALGORITHM

The complete procedure of our method with fine-tuning is outlined in Algorithm 1. Given an input graph $G = (A, X, Y)$, we employ a subsampling function $\mathcal{T}$ to sample subgraphs for each node. Subsequently, we generate the condition $P_i'$ using node attributes and positional embeddings $P_i$ using adjacency matrix for each subgraph $G_i$. The training dataloader $\mathcal{S}$ is then created with a batch size of 128 for subgraphs of training nodes. During each iteration, batched graphs are inputted into the frozen pre-trained model $g_\theta^\star$, while the condition is fed into the trainable copy $g_{\theta_c}$. These two components are interconnected using zero MLPs. Finally, the representations $H$ are passed through a classifier $p_\omega$, with the cross-entropy loss $\mathcal{L}$ utilized to compute the classification error $\ell_{\text{sup}}$. The parameters of the trainable copy, zero MLPs, and classifier are optimized by minimizing the loss.

## B  TIME COMPLEXITY ANALYSIS

Given an sparse input graph $G = (A, X, X_{\text{PE}}, Y)$, the attribute matrix $X \in \mathbb{R}^{N \times D}$, the positional embedding $X_{\text{PE}} \in \mathbb{R}^{N \times K}$ where $K \ll D$. Let the hidden size and the number of layers in the model be denoted as $H$ and $L$, respectively. $H$ is comparable to $D$ in most cases, so we consider them to be the same in this analysis for simplicity.

The time complexities of the baselines and our method are outlined in Table 5. Notably, GIN's input features comprise $D$ dimensions, whereas GCC's input dimension, denoted as $K$, is considerably smaller than $D$. Consequently, GCC exhibits higher efficiency compared to GIN. Our method involves both the frozen GCC and its trainable counterpart. The main computational workload is twofold compared to GCC, encompassing the additional processing time required for zero MLPs and feature summations.

**Table 5: Time complexity analysis of baselines and our method.**

| Data | Method | Forward Time complexity |
|---|---|---|
| A,X
A,$X_{\text{PE}}$ | GIN
GCC | $O(LED + LND^2)$
$O(LED + (L-1)ND^2 + NKD)$ |
| A,$X_{\text{PE}}$,X | GCC+GraphControl | $O(2(LED + (L-1)ND^2 + NKD)$
$+ NK + NK^2 + ND + ND^2)$ |

## C  EXPERIMENT

In this section, we will provide detailed information about experiments. Firstly, we introduce the datasets used in the main content in detail. And then we introduce the baselines used in the main content. Lastly, we provide the hyper-parameters of experiments.

### C.1  Datasets

*C.1.1  Pretraining datasets.* The pre-training datasets utilized by GCC are outlined in Table 6. These datasets fall into two main categories: academic graphs, including Academia, and two DBLP datasets, and social graphs, including IMDB, Facebook, and LiveJournal datasets. The Academia dataset is sourced from NetRep [32], and the two DBLP datasets are obtained from SNAP [1] and NetRep [32] respectively. Additionally, the IMDB and Facebook datasets are gathered from NetRep [32], and the LiveJournal dataset is collected from SNAP [1].

*C.1.2  Downstream datasets.* The datasets can be categorized into two groups: attributed datasets (Cora_ML, Amazon Photo, DBLP, and Coauthor Physics) and non-attributed datasets (USA Airport, Europe Airport, Brazil Airport, and H-index). Below are detailed descriptions of these datasets.

- Amazon Photo [36] consists of segments from the Amazon co-purchase graph [22]. In this dataset, nodes represent goods, edges signify frequent co-purchases between goods, node features are bag-of-words encoded product reviews, and class labels are assigned based on product categories.
- Cora_ML and DBLP datasets [3] are citation networks used for predicting article subject categories. In these datasets, graphs are created from computer science article citation links. Nodes represent articles, and undirected edges signify citation links between articles. Class labels are assigned based on paper topics.
- In the Coauthor Physics dataset [36], graphs are co-authorship networks derived from the Microsoft Academic Graph. Nodes in this dataset represent authors and are connected by edges if they co-authored a paper. Node features

**Table 6: Statistics of pre-training datasets.**

| Dataset | Academia | DBLP (SNAP) | DBLP (NetRep) | IMDB | Facebook | LiveJournal |
|---|---|---|---|---|---|---|
| #Nodes | 137,969 | 317,080 | 540,486 | 896,305 | 3,097,165 | 4,843,953 |
| #Edges | 739,384 | 2,099,732 | 30,491,458 | 7,564,894 | 47,334,788 | 85,691,368 |

**Table 7: Hyper-parameters for GIN(A,X) baseline.**

| | Cora_ML | Amazon-Photo | DBLP | Coauthor-Physics | USA | Europe | Brazil | H-index |
|---|---|---|---|---|---|---|---|---|
| Model | GIN | GIN | GIN | GIN | GIN | GIN | GIN | GIN |
| # Hidden size | 64 | 64 | 64 | 64 | 64 | 64 | 64 | 64 |
| # Layers | 4 | 4 | 4 | 4 | 4 | 4 | 4 | 4 |
| # Epochs | 1000 | 800 | 100 | 100 | 100 | 100 | 200 | 200 |
| Learning rate | 1e-3 | 1e-2 | 1e-3 | 1e-3 | 1e-3 | 1e-2 | 1e-2 | 1e-3 |
| Optimizer | Adam | Adam | Adam | Adam | Adam | Adam | Adam | Adam |
| Weight decay | 5e-4 | 5e-4 | 5e-4 | 5e-4 | 5e-4 | 5e-4 | 5e-4 | 5e-4 |

represent paper keywords from each author's publications, and class labels indicate the authors' most active fields of study.

- The USA Airport dataset [30] consists of data collected from the Bureau of Transportation Statistics[4] between January and October 2016. The network comprises 1,190 nodes and 13,599 edges, with a diameter of 8. Airport activity is quantified by the total number of people who passed through the airport (both arrivals and departures) during the corresponding period.
- The Europe Airport dataset [30] comprises data gathered from the Statistical Office of the European Union (Eurostat)[5] between January and November 2016. The network consists of 399 nodes and 5,995 edges, with a diameter of 5. Airport activity is evaluated based on the total number of landings and takeoffs during the corresponding period.
- The Brazil Airport dataset [30] is sourced from the National Civil Aviation Agency (ANAC)[6] and covers the period from January to December 2016. The network comprises 131 nodes and 1,038 edges, with a diameter of 5. Airport activity is quantified based on the total number of landings and takeoffs during the corresponding year.
- The H-index dataset [29] is derived from a co-authorship graph extracted from OAG[50]. To enhance suitability for the node classification task, smaller subgraphs are extracted from the original graph due to its vast scale. This resulting network comprises 5,000 nodes and 44,020 edges, with a diameter of 7. Labels in the H-index dataset indicate whether the author's h-index is above or below the median.

## C.2 Hyper-parameters

In this section, we will provide the hyper-parameters used in our experiments. Table 7 lists the parameters of baselines. And Table 8 lists

the parameters of structural pre-training methods. Lastly, Table 9 provides the details of transfer learning.

## C.3 Baselines

In Section 5.3, four pre-training methods are incorporated: GCC, GRACE, simGRACE, RoSA, and COSTA. In this section, we will elucidate these methods.

- GCC [29] is a structural pre-training method based on local structural information. It utilizes position embeddings as model input to learn transferable structural patterns through subgraph discrimination.
- GRACE [57] is node-node graph contrastive learning method. It designs two augmentation functions (*i.e.*, removing edges and masking node features) to generate two augmented views. Then a shared graph model will be applied on augmented views to generate node embedding matrices. The node representations augmented from the same original node are regarded as positive pairs, otherwise are negative pairs. Lastly, pairwise loss (*e.g.*, InfoNCE [27]) will be applied on these node matrices.
- simGRACE [44] eliminates data augmentation while introducing encoder perturbations to generate distinct views for graph contrastive learning.
- RoSA [55] is a robust self-aligned graph contrastive framework which does not require the explicit alignment of nodes in the positive pairs so that allows more flexible graph augmentation. It proposes the graph earth move distance (g-EMD) to calculate the distance between unaligned views to achieve self-alignment. Furthermore, it will use adversarial training to realize robust alignment.
- COSTA [52] proposes feature augmentation to decrease the bias introduced by graph augmentation.

Received 20 February 2007; revised 12 March 2009; accepted 5 June 2009

---

[4] https://transtats.bts.gov/
[5] http://ec.europa.eu/
[6] http://www.anac.gov.br/

**Table 8: Hyper-parameters for pre-training method GRACE.**

|  | Cora_ML | Amazon-Photo | DBLP | Coauthor-Physics | USA | Europe | Brazil | H-index |
|---|---|---|---|---|---|---|---|---|
| Model | GIN | GIN | GIN | GIN | GIN | GIN | GIN | GIN |
| # Hidden size | 64 | 64 | 64 | 64 | 64 | 64 | 64 | 64 |
| # Layers | 4 | 4 | 4 | 4 | 4 | 4 | 4 | 4 |
| # Epochs | 20 | 50 | 100 | 20 | 50 | 500 | 200 | 100 |
| Learning rate | 1e-3 | 1e-4 | 1e-3 | 1e-4 | 1e-3 | 1e-2 | 1e-2 | 1e-3 |
| Optimizer | Adam | Adam | Adam | Adam | Adam | Adam | Adam | Adam |
| Weight decay | 5e-4 | 5e-4 | 5e-4 | 5e-4 | 5e-4 | 5e-4 | 5e-4 | 5e-4 |
| Walk steps | 256 | 256 | 256 | 256 | 256 | 256 | 256 | 256 |
| Restart rate | 0.3 | 0.5 | 0.3 | 0.5 | 0.3 | 0.5 | 0.5 | 0.5 |
| $\tau$ | 0.2 | 0.2 | 0.2 | 0.2 | 0.2 | 0.2 | 0.2 | 0.2 |
| $p_{f,1}$ | 0.2 | 0.2 | 0.2 | 0.2 | 0.2 | 0.2 | 0.3 | 0.2 |
| $p_{f,2}$ | 0.3 | 0.3 | 0.3 | 0.3 | 0.3 | 0.3 | 0.2 | 0.3 |
| $p_{e,1}$ | 0.2 | 0.2 | 0.2 | 0.2 | 0.2 | 0.2 | 0.2 | 0.2 |
| $p_{e,1}$ | 0.3 | 0.3 | 0.3 | 0.3 | 0.3 | 0.3 | 0.3 | 0.3 |

**Table 9: Hyper-parameters for Transfer Learning (GraphControl with GCC pre-trained model).**

|  | Cora_ML | Amazon-Photo | DBLP | Coauthor-Physics | USA | Europe | Brazil | H-index |
|---|---|---|---|---|---|---|---|---|
| Model | GCC | GIN | GCC | GCC | GCC | GCC | GCC | GCC |
| # Hidden size | 64 | 64 | 64 | 64 | 64 | 64 | 64 | 64 |
| # Layers | 4 | 4 | 4 | 4 | 4 | 4 | 4 | 4 |
| # Epochs | 100 | 100 | 100 | 100 | 100 | 100 | 400 | 100 |
| Learning rate | 0.5 | 0.5 | 0.1 | 0.01 | 0.3 | 0.2 | 0.1 | 0.1 |
| Optimizer | AdamW | AdamW | Adam | Adam | SGD | SGD | SGD | SGD |
| Weight decay | 5e-4 | 5e-4 | 5e-4 | 1e-2 | 1e-3 | 5e-4 | 1e-3 | 5e-4 |
| Walk steps | 256 | 256 | 256 | 256 | 256 | 256 | 256 | 256 |
| Restart rate | 0.8 | 0.8 | 0.8 | 0.8 | 0.5 | 0.5 | 0.3 | 0.5 |
| Threshold | 0.17 | 0.2 | 0.3 | 0.15 | 0.15 | 0.15 | 0.3 | 0.17 |

**Table 10: Hyper-parameters for Domain Transfer (GraphControl with other pre-trained models).**

|  | Cora_ML | Amazon-Photo | DBLP | Coauthor-Physics | USA | Europe | Brazil | H-index |
|---|---|---|---|---|---|---|---|---|
| Model | GIN | GIN | GIN | GIN | GIN | GIN | GIN | GIN |
| # Hidden size | 64 | 64 | 64 | 64 | 64 | 64 | 64 | 64 |
| # Layers | 4 | 4 | 4 | 4 | 4 | 4 | 4 | 4 |
| # Epochs | 100 | 100 | 100 | 100 | 100 | 100 | 200 | 200 |
| Learning rate | 1e-1 | 1e-3 | 1e-3 | 1e-3 | 1e-3 | 1e-3 | 1e-3 | 5e-4 |
| Optimizer | Adam | Adam | Adam | Adam | Adam | Adam | Adam | SGD |
| Weight decay | 1e-3 | 5e-4 | 5e-4 | 5e-4 | 5e-4 | 5e-4 | 5e-4 | 5e-4 |
| Walk steps | 256 | 256 | 256 | 256 | 256 | 256 | 256 | 256 |
| Restart rate | 0.3 | 0.5 | 0.3 | 0.3 | 0.3 | 0.3 | 0.3 | 0.3 |

