# OpenReview forum: "GraphControl: Adding Conditional Control to Universal Graph Pre-trained Models for Graph Domain Transfer Learning"
_ACM.org/TheWebConf/2024/Conference — TheWebConf24 Oral_

### Official Review · Reviewer_Ftuk · 2023-10-25

**Novelty:** 5
**Technical Quality:** 5

**Review:**

This paper addresses the challenge of transferring pre-trained graph models to web applications like paper classification and website recommendation. While pre-trained models on unlabeled graph data offer potential for downstream tasks, differences in attribute semantics across graphs create a "transferability-specificity dilemma." To address this, the paper introduces GraphControl, inspired by ControlNet. GraphControl aligns input spaces across graphs and incorporates target data's unique characteristics as conditional inputs, enabling the utilization of downstream data statistics. Extensive experiments demonstrate substantial performance improvements, achieving 2-3x gains on datasets like Cora_ML and Amazon-Photo and outperforming training-from-scratch methods by over 5% on some datasets.

**Questions:**

1. Since the paper draws inspiration from the image-text diffusion, there should be one section discuss about those algorithms rather than a short introduction in the method section.

2. The datasets utilized are quite small, especially the link prediction one. I would suggest authors conducting experiments on larger datasets.

3. There is a lack of time and memory complexity analysis.

4. It seems that the Analysis of subsampling hyperparameters on Cora_ML. I think the paper could be better with further explanation.

5. The pretraining dataset is still not large enough in current work

**Reviewer Confidence:**

3: The reviewer is confident but not certain that the evaluation is correct

**Scope:**

2: The connection to the Web is incidental, e.g., use of Web data or API

---

### Official Review · Reviewer_M5WR · 2023-11-20

**Novelty:** 4
**Technical Quality:** 6

**Review:**

This paper introduces GraphControl, a deployment module aimed at addressing the challenges of graph transfer learning.  It utilizes a kernel matrix to measure the graph attribute similarities between graph attributes and introduce the idea of control-net to incorporate the information as input conditions for graph transfer learning.  The experiments demonstrate that the proposed algorithm outperforms existing baselines in scenarios of graph fine-tuning and prompt tuning.  The methodology is straightforward and is clearly articulated. The paper presents a comprehensive set of experiments, and the analysis is discussed thoroughly.
Strengths:
1.The presentation of the paper is coherent, and the overall technical quality is commendable.
The experimental work is substantial, and the discussions provided are thorough and informative.
Weaknesses:
1.The paper employs well-established methods in transfer learning, but it appears to lack significant innovative contributions.
2.In the experimental part, for the baseline graph pre-training methods designed for attributed graphs, the authors perform pre-training on structural information, omitting node attributes. This approach may not provide a fair comparison to these baselines, considering the considerable differences in outcomes.
3.It would be beneficial to include additional baselines for attributed graphs that utilize both structural and attribute information (A, X) in Table 2, other than merely GIN, to more effectively showcase the superiority of the proposed transfer learning method in comparison to conventional approaches.
4.The manuscript could be improved by addressing the inconsistencies found in certain mathematical notations, such as those pertaining to positional embeddings.

**Questions:**

1.Could the authors provide a more equitable experimental setup for benchmarking against baseline methods? If the current experimental conditions are deemed fair, an elucidation of the same is warranted.
2.Comparisons with other state-of-the-art graph transfer learning methods and attributed graph algorithms would be invaluable.

**Reviewer Confidence:**

4: The reviewer is certain that the evaluation is correct and very familiar with the relevant literature

**Scope:**

4: The work is relevant to the Web and to the track, and is of broad interest to the community

---

### Official Review · Reviewer_XhfM · 2023-11-26

**Novelty:** 4
**Technical Quality:** 5

**Review:**

What is this paper about：
This paper aims to address challenges related to the transferability of pre-trained graph models to downstream tasks, specifically in the context of graph-structured data. The authors introduce a solution called GraphControl, inspired by ControlNet, to overcome the "transferability-specificity dilemma."

what contributions does it make:
1.The authors propose an innovative deployment module, GraphControl. This module improves graph domain transfer learning by aligning the input space across various graphs and incorporating unique characteristics of target data as conditional inputs.
2.The authors design a condition generation module and the ablation study in the experiment reflects its necessity.
3.The proposed method enhances the adaptability of pre-trained models on target attributed datasets, achieving a performance gain of 1.4-3x.

the main strengths:
1.The generated conditions are progressively integrated into the model during fine-tuning or prompt tuning through ControlNet.
2.The proposed method is highly flexible and can be seamlessly integrated with a variety of existing pre-trained models.
3.The model outperforms training-from-scratch methods on target data with a comparable margin and exhibits faster convergence.
the main weaknesses:
1.Figure 1 lacks any form of analysis or interpretation.
2.The experimental analysis needs to be more detailed, such as why the improvement of non-attributed datasets is smaller than the effect of attributed datasets?

**Questions:**

According to the complexity analysis in the appendix, we can see that the method proposed in this paper is more complicated than GCC. How much longer is the running time? If the increase is not much, it can mean that the method in this paper is efficient, but if it is much higher, it may not be very effective.

**Reviewer Confidence:**

3: The reviewer is confident but not certain that the evaluation is correct

**Scope:**

4: The work is relevant to the Web and to the track, and is of broad interest to the community

---

### Official Review · Reviewer_nVcU · 2023-11-28

**Novelty:** 6
**Technical Quality:** 6

**Review:**

**Quality**:

The proposed innovative GraphControl module aims to resolve the "transferability-specificity dilemma" in graph transfer learning. It utilizes universal structural pre-trained models and incorporates unique features of downstream data as input conditions. This approach effectively integrates valuable downstream attributes while safeguarding against detrimental noise during fine-tuning.

**Pros:**

1. Introduction of the GraphControl module to tackle the "transferability-specificity dilemma."

2. Designing a condition generation module to preprocess downstream-specific information for effective integration into pre-trained models.

3. Extensive experiments demonstrating the significant enhancement of adaptability of pre-trained models on downstream datasets, showcasing substantial performance gains compared to training-from-scratch methods.

4. This innovative approach shows promise for more effective deployment of pre-trained models in real-world web applications, facilitating enhanced adaptability and performance on diverse datasets.

5. This work can be easily integrated with recent work on graph cue learning, pushing for further completeness of the graph pre-training paradigm.

6. The paper is well-written and easy to understand.

**Questions:**

Q1: Another recent and promising trend in constructing foundational graph models involves textual graphs, as seen in [1], [2]. Could the authors elaborate on the potential for integrating GraphControl with these advancements?

**Reference:**

[1] Liu H, Feng J, Kong L, et al. One for All: Towards Training One Graph Model for All Classification Tasks[J]. arXiv preprint arXiv:2310.00149, 2023.

[2] He X, Bresson X, Laurent T, et al. Explanations as Features: LLM-Based Features for Text-Attributed Graphs[J]. arXiv preprint arXiv:2305.19523, 2023.

**Reviewer Confidence:**

4: The reviewer is certain that the evaluation is correct and very familiar with the relevant literature

**Scope:**

4: The work is relevant to the Web and to the track, and is of broad interest to the community

---

### Decision · Program_Chairs · 2024-01-22

**Decision:**

Accept (Oral)

**Comment:**

The paper introduces GraphControl, a module designed to enhance the transferability of graph pre-trained models to different domains by resolving the "transferability-specificity dilemma". This is achieved by aligning input spaces across various graphs and incorporating unique characteristics of target data as conditional inputs. The paper is well-structured and presents an extensive set of experiments demonstrating substantial performance improvements in graph domain transfer learning.

 The reviewers have predominantly positive sentiments about the paper. They commend the novelty and technical quality of the work, highlighting its clear structure and extensive experimental validation. There are concerns regarding the clarity in the presentation of certain aspects, such as the time and memory complexity analysis, and the need for experiments on larger and more diverse datasets.


 Overall it's an interesting and timely paper for graph ML community, and I recommend the acceptance of this paper for oral presentation. The authors should consider incorporating the suggested improvements in their final version.